# Dark Matter as Variations in the Electromagnetic Zero-Point Field Induced by Baryonic Matter

**Yehonatan Knoll**

Independent Researcher, Haifa 3254605, Israel; yonatan2806@gmail.com

**Abstract:** Cold dark-matter, as a solution to the so-called dark-matter problem, suffers from a major internal conflict: In order to dodge direct detection for so long, it must have an unobservably small (non gravitational) interaction with mundane matter, and yet it manages to 'conspire' with it such that, in single galaxies, its distribution can be inferred from that of mundane matter via the MOND phenomenology. This conflict is avoided if the missing, transparent component of the energy-momentum tensor is due to variations in some electromagnetic 'zero point field' (ZPF) which is sourced by mundane matter and contains both its advanced and retarded fields. The existence of a ZPF thus modulated by mundane matter, follows from a proper solution to the self-force problem of classical electrodynamics (CED), recently proposed by the author, which renders CED compatible with the statistical predictions of QM. The possibility that 'dark matter' is yet another, hitherto ignored facet of good-old classical electrodynamics, therefore seems no less plausible than it being a highly exotic and conspirative new form of matter. Tests for deciding between the two are proposed.

**Keywords:** dark-matter problem; foundations of quantum mechanics; foundations of classical electrodynamics

## 1. Introduction

At the turn of the twentieth century, classical electrodynamics (CED) was the only game in town. Following Einstein's resolution of its Galilean non-covariance problem, one could have thought that a theory of everything was just around the corner. And yet, to paraphrase Kelvin, a few dark clouds hovered over CED:

1. Freely moving charges in a lab trace parabolas rather than straight lines. CED needed Newton's gravity by its side, with its distinct (Galilei rather than Lorentz) symmetry group, making it impossible to merge the two into a consistent theory.

2. CED was mathematically ill-defined due to the so-called classical self-force problem: Both the Lorentz force equation of a point charge and the total energy of a group of interacting point charges are ill-defined [1].

3. CED was not generally covariant. The coordinates appearing in CED's Minkowskian form as well as the dynamical fields are merely abstractions of physical entities—rods, clocks, magnetometers, etc.; spacetime is not "marked" with coordinates, nor are there "little vectors" scattered in it. A fundamental theory, in turn, should be able to represent any such physical entity, and if the required mathematical representation resorts to coordinates, an infinite regression (or circularity) is created. The only way to avoid it is for coordinates to enter a description of nature as "scaffolding", used in calculating the "real thing": a measurement, which is just some coordinate-independent number (e.g., the number of clock ticks). A particularly simple way of guaranteeing the scaffolding independence of the real thing is to make CED's equations look the same in any coordinate system, identifying the results of measurements with certain

(coordinate-independent) scalars. The principle of general covariance, which crept into physics as a mathematical corollary of Einstein's field equations, could have therefore been proclaimed much earlier.

4.　CED began showing some discrepancies with observations, such as the photoelectric effect and black-body radiation, with no apparent resolution in sight.

In 1905, therefore, CED was no more than a rough sketch, or first draft of a theory, certainly not a mature one. It worked so well despite its internal inconsistencies simply because it was tested in a rather limited domain, where ad hoc "cheats" enabled the extraction of definite results from an ill-defined, conceptually flawed mathematical apparatus. When the domain of CED was subsequently extended and no cheating method would lead to the experimental result anymore, the demise of CED began and alternatives sprung. In the current paper, we argue that seeking alternatives to a successful proto-theory is a bad methodology; physicists at the first quarter of the twentieth century should have first elevated CED to the status of a theory, preserving those of its features responsible for its success, and only then figured what else, if anything, was needed in physics.

And such an elevation is possible: a solution to the non-covariance problem can begin with the standard procedure of expressing differential equations in curvilinear coordinates, $\xi^\mu$. Given CED's Minkowskian form in coordinates $x^\mu$ (experimentally found to "work" in some freely falling "labs"), each new coordinate system introduces a symmetric transformation matrix:

$$g_{\mu\nu} = \frac{\partial x^\alpha}{\partial \xi^\mu} \frac{\partial x^\beta}{\partial \xi^\nu} \eta_{\alpha\beta}, \quad \eta = \text{diag}(1, -1, -1, -1),$$

completely encoding the effect of the transformation. The geodesic equation then becomes just the Lorentz force equation in empty space, expressed in curvilinear coordinates. However, $g_{\mu\nu}$—10 independent functions—is an infinite set of parameters, changing from one coordinate system to another, which is exactly the definition of an equation not being covariant with respect to a group of transformations, and the standard way of coping with such non-covariance is to elevate the status of those parameters to that of dynamical variables. Further recalling that, by its definition, $g_{\mu\nu}$ transforms as a second rank tensor, the simplest nontrivial covariant choice for the equation to be satisfied by $g_{\mu\nu}$ is Einstein's field equations:

$$AR_{\mu\nu} + Bg_{\mu\nu}R + Cg_{\mu\nu} = P_{\mu\nu}, \tag{1}$$

with $R_{\mu\nu}$ and $R$ as the once- and twice-contracted Riemann tensor; $P_{\mu\nu}$ as the total energy-momentum tensor of matter+radiation; and $A, B$, and $C$ as some constants to be determined by observations and mathematical consistency (Between 1915 and 1919, Einstein himself had already proposed three different sets of constants [2].). Note the emphasis on general covariance rather than geometry and, in particular, that $g_{\mu\nu}$ has no a priori metrical meaning. This alternative approach is, of course, much easier to adopt in hindsight, but the point stands: Not only special relativity is buried in CED (as attested by the title of Einstein's first paper on relativity) but also general relativity (GR). A solution to problem 1 is therefore a corollary of the solution to 3, namely, CED + gravity is just generally covariant CED.

In the final chapter of his classic treatise on CED [3] p. 745, Jackson addresses problem 2—the self-force problem: "Why is it that we have waited so long?" He apologetically asks and answers: "a completely satisfactory classical treatment of the reactive effect of radiation does not exist. The difficulties presented by this problem touch one of the most fundamental aspects of physics, the nature of an elementary particle.". A recently proposed novel mathematical construction, dubbed extended charge dynamics (ECD), first appearing in [4], then fine-tuned in [1], and briefly discussed in Section 2, provides a proper solution to the classical self-force problem, and, indeed, touches "the [very] nature of an elementary particle". By "proper", we mean a mathematically well-defined realization of the *basic tenets of CED* which are responsible for its immense success: Maxwell's equations and local energy-momentum (e-m) conservation.

There remains problem 4. In [5], it was shown that a proper solution to 1–3, namely generally covariant ECD, leads to a new problem: statistical aspects of ensembles of ECD solutions, such as those involved in a scattering experiment, cannot be read from ECD alone, requiring a complementary statistical theory. It is argued there that quantum mechanics is that missing complementary statistical theory, which solves problem 4. With the advent of QM, the associated conceptual difficulties became an issue also in astronomy: it is no longer clear what to put on the right-hand-side (r.h.s.) of (1) in the first place. ECD's resolution of those difficulties imply, among else, that no approximation is involved in using the classical e-m tensor on the r.h.s. of (1).

With CED's original four problems apparently solved, we fast-forward the evolution of twentieth century physics, reviewing it in the new light shed by ECD. In Section 3, dealing with particle physics, we briefly sketch the results of [5] regarding the so-called block-universe (BU) view, mandated by both SR and GR. A clear distinction is drawn between the (classical) *ontology* of the BU, allegedly ECD, and various statistical descriptions thereof, such as the standard model of particle physics. Section 3.1, presenting a tentative model of matter based solely on ECD, is not crucial for the understanding of the rest of the paper and may be skipped on first reading. It is included, nonetheless, because the proposed 'transparent matter' is not yet another placeholder for 'something', only with a vanishing trace for its energy-momentum tensor. Instead, it (allegedly) emerges as an integral part of the representation of ordinary matter, hence the digression into particle physics. Along the way, simple explanations are provided to persistent mysteries in particle physics, such as the quantization of the electric charge. In Section 4, we get to the crux of our hypothesis: that the missing, transparent component of the energy-momentum tensor is due to modulations of the intensity of the ECD zero-point field induced by the surrounding baryonic matter. That zero-point field has unique properties, distinguished from the field by the same name in stocahstic electrodynamics and, as should be clear from Section 3, from the "quantum vacuum". The "transparent matter" model developed there is only preliminary but still generates distinct testable predictions. Its radical implications for cosmology are discussed in [6].

Finally, a note regarding the broader context of the paper: For the past eighty years or so, progress in physics consisted mostly of a series of "epicycles", each added in response to a discordant observation. This natural process, enjoying the merit of "backward compatibility", can either continue forever or else stagnate, as the task of adding an epicycle becomes harder due to an expanding experimental body of knowledge. Those believing that the latter scenario had occurred, hence that the time is ripe to consider a paradigm shift, are still a minority among physicists, but their number is steadily increasing and for good reasons. Now, the problem with a paper advocating a paradigm shift is that it would be futile to zoom-in on an isolated patch of the big picture; one's proposal could elegantly solve a conundrum in one domain but clash with observations in another or even lack extensions thereto (MOND being such an example; the entire program of particle physics, explaining but a tiny fraction of the alleged matter "out there", is to a large extent another). Instead, it has to depict an alternative panoramic picture, hopefully convincing that a genuine landscape could lie behind it. The reader is therefore warned that, given obvious resource limitations, the picture he/she is about to see is, in part, of low resolution compared with the norm adhered to in standard, domain-specific scientific publications.

## 2. Extended Charge Dynamics (ECD) in Brief

First appearing in [4] and then fine-tuned and related to the self-force problem in [1], ECD is a concrete realization of the two obvious pillars of classical electrodynamics (CED) referred to as the *basic tenets of CED* , which are Maxwell's equations in the presence of a conserved source—the antisymmetry of $F$ implies current conservation—due to all particles (labelled by $a$)

$$\partial_\nu F^{\nu\mu} \equiv \partial^\nu \partial_\nu A^\mu - \partial^\mu \partial_\nu A^\nu = \sum_a j^{(a)\,\mu} \,, \tag{2}$$

$$\partial_\mu j^{(a)\,\mu} = 0 \,, \tag{3}$$

with $F_{\mu\nu} = \partial_\mu A_\nu - \partial_\nu A_\mu$ as the antisymmetric Faraday tensor, and local "Lorentz force equation"

$$\partial_\nu T^{(a)\,\nu\mu} = F^{\mu\nu} j^{(a)}{}_\nu \,, \tag{4}$$

with $T^{(a)}$ as the symmetrical "matter" e-m tensor associated with particle $a$. Defining the *canonical tensor*

$$\Theta^{\nu\mu} = \frac{1}{4} g^{\nu\mu} F^{\rho\lambda} F_{\rho\lambda} + F^{\nu\rho} F_\rho{}^\mu \,, \tag{5}$$

we get from (2) and (5) Poynting's theorem:

$$\partial_\nu \Theta^{\nu\mu} = -F^\mu{}_\nu \sum_a j^{(a)\,\nu} \,. \tag{6}$$

Summing (4) over $a$ and adding to (6), we get local e-m conservation

$$P := \Theta + \sum_a T^{(a)} \quad \Rightarrow \quad \partial_\nu P^{\nu\mu} = 0 \,, \tag{7}$$

and, purely by the symmetry and conservation of $P^{\nu\mu}$, also generalized angular momentum conservation

$$\partial_\mu \mathcal{J}^{\mu\nu\rho} = 0 \,, \quad \mathcal{J}^{\mu\nu\rho} = \epsilon^{\nu\rho\lambda\sigma} P^\mu{}_\sigma x_\lambda \,. \tag{8}$$

As shown in [1], for $j^{(a)}$ and $T^{(a)}$ co-supported on a common world-line corresponding to "point-particle" CED, no realization of the basic tenets exists. Their ECD realization therefore involves $j$ and $T$ extending beyond this line support yet still localized about it, representing what can be called "extended particles" with non-rigid internal structures. Nevertheless, the reader must not take too literally this name, as both $j$ and $T$ associated with distinct particles are allowed to overlap or cross one another, which is a critical point in our subsequent analysis. Moreover, the magnetic dipole moment and the angular momentum associated with a single spin-$\frac{1}{2}$ ECD particle at rest have a fixed nonvanishing value which cannot be "turned off", viz. that particle is not some "rotating, electrically charged liquid drop" eventually dissipating its angular momentum and magnetic dipole. Finally, it is stressed that the ECD objects carrying a particle label, such as $j^{(a)}$ and $T^{(a)}$, collectively dubbed *particle densities*, should not be viewed as time-varying three dimensional extended distributions but, rather, as covariant four dimensional "extended world-lines". This point, too, is critical.

As shown in appendix D of [1], when a charged body moves in a weak external electromagnetic field which is slowly varying over the extent of the body, a coarse description of its path is given by solutions of the Lorentz force equation in that field. This is a direct consequence of the basic tenets, hence the name "local Lorentz force equation" given to (4). In the presence of a strong or rapidly varying external field, however, a general ECD solution, whether representing a single (elementary-) particle or a bound aggregate thereof (composite particle), has additional attributes besides its average position in space, facilitated by its extended structure, and moreover, even its coarse path could deviate substantially from the Lorentz force law. In particular, ECD paths could look like those depicted in Figure 1a.

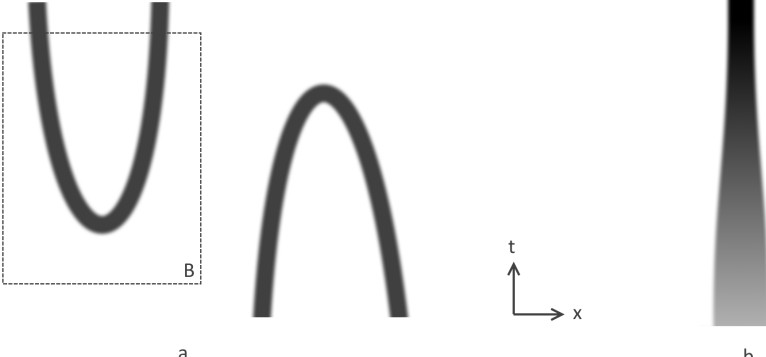

**Figure 1.** Nonclassical scenarios for extended charge dynamics (ECD) particles: (**a**) creation and annihilation of a pair and (**b**) scale transition (the varying gray-level represents charge density)

Applying Stoke's theorem to local charge conservation (3) and box B in Figure 1a, we see that the two created/annihilated particles must be of opposite charges. However, the reader should not rush to a conclusion that those are a particle-antiparticle pair despite ECD's "CPT" symmetry:

$$A(x) \mapsto -A(-x), \quad j(x) \mapsto -j(-x), \quad T(x) \mapsto T(-x) \quad \Longrightarrow \tag{9}$$
$$P(x) \mapsto P(-x), \quad \mathcal{J}(x) \mapsto -\mathcal{J}(-x).$$

It is only when the two "branches" are sufficiently removed from each other and have attained some metastable state that a particle-type label can be assigned to them, and it may very well be that this never happens; either branch could end up being part of a composite particle before stabilizing. *This offers a particularly simple explanation for the observed imbalance between matter and antimatter.*

Applying Stoke's theorem to e-m conservation (7) and box B, we further see that the disappearance/emergence of mechanical e-m must be balanced by either a corresponding release/absorption of electromagnetic e-m or by the creation/annihilation of another pair (or pairs).

## 2.1. Advanced Solutions of Maxwell'S Equations

In a universe in which no particles implies no electromagnetic field (viz., $j \equiv 0 \Rightarrow A =$ pure gauge) the most general (sensible, Lorentz and gauge covariant) solution to Maxwell's equations (2), takes the form

$$A^{\mu}(x) = \int d^4 x' \left[ \alpha_{\text{ret}}(x') K_{\text{ret}}{}^{\mu\nu}(x - x') + \alpha_{\text{adv}}(x') K_{\text{adv}}{}^{\mu\nu}(x - x') \right] j_{\nu}(x'), \tag{10}$$

for some (Lorentz invariant) spacetime-dependent $\alpha$'s, constrained by $\alpha_{\text{ret}} + \alpha_{\text{adv}} \equiv 1$, where $K_{\text{ret} \atop \text{adv}}$ is the advanced and retarded Green's function of (2), defined by

$$\left( g_{\mu\nu}\partial^2 - \partial_{\mu}\partial_{\nu} \right) K_{\text{ret} \atop \text{adv}}{}^{\nu\lambda}(x) = g_{\mu}{}^{\lambda}\delta^{(4)}(x), \tag{11}$$

$$K_{\text{ret} \atop \text{adv}}(x) = 0 \quad \text{for } x^0 \lessgtr 0, \tag{12}$$

(More accurately, (11) and (12) do not uniquely define $K$, but the remaining freedom can be shown to translate via (10) to a gauge transformation $A \mapsto A + \partial\Lambda$, consistent with the gauge covariance of ECD). The standard proviso added to CED, $\alpha_{\text{adv}} \equiv 0$, which is not implied by the observed arrow-of-time [1,5], is moreover incompatible with ECD. In other words, one cannot impose a choice of $\alpha$'s on ECD currents but, instead, must read the choice from a global consistent solution, involving fields and currents.

The asymmetry between advanced and retarded solutions, manifested in the arrow-of-time (AOT) is intimately related to ECD's CPT symmetry (9). That is, for our universe to have an oppositely pointing AOT, it would also need to have every particle replaced with its antiparticle. The role of

cosmology in this context is discussed in [6]. We shall further return to the AOT in Section 3.1.4 dealing with the explanation given by ECD to photon-related phenomena.

## 2.2. Scale Covariance

Scale covariance is a symmetry of a physical theory in Minkowski's spacetime (not to be confused with Weyl invariance), and it is just as natural a symmetry as translation covariance. A fundamental description of nature should not include a privileged length scale, just like it should not include a privileged position—both ought to appear as attributes of specific *solutions* rather than of the equations. This is not only an aesthetically appealing demand. Extrapolating the very little we know about nature to scales much larger or smaller than our native scale is groundless unless our description of nature is scale covariant.

ECD is scale covariant by virtue of its symmetry

$$A(x) \mapsto \lambda^{-1} A\left(\lambda^{-1}x\right), \quad j(x) \mapsto \lambda^{-3} j\left(\lambda^{-1}x\right), \quad T(x) \mapsto \lambda^{-4} T\left(\lambda^{-1}x\right) \tag{13}$$
$$\implies \Theta(x) \mapsto \lambda^{-4} \Theta\left(\lambda^{-1}x\right), \quad P(x) \mapsto \lambda^{-4} P\left(\lambda^{-1}x\right), \quad \mathcal{J}(x) \mapsto \lambda^{-3} \mathcal{J}\left(\lambda^{-1}x\right),$$

with the two free parameters of ECD unchanged. The exponent of $\lambda$ is referred to as the *scaling dimension* of a density; hence, by definition, it is 0 for those two ECD parameters. The scale factor, $\lambda$, which in the present context is taken to be positive, can in fact be an arbitrary nonvanishing real number thereby merging scaling symmetry with CPT symmetry (9).

ECD, however, takes scale covariance one step beyond the formal symmetry (13) (cf. Sections 1.2 and 2 Extended in [4] dealing with scale covariance of point-particle CED). ECD particles can *dynamically* undergo a scale transformation, as illustrated in Figure 1b. This is not some curiosity but rather a generic feature of any scale covariant theory in which particles acquire their observed mass dynamically. Now, as can be expected, a formally conserved Noether current associated with symmetry (13) exists for ECD, implying, among others, that the scale/mass of a particle is conserved, which conflicts with our assumption that particles can "move in scale". Remarkably, the associated ECD charge can "leak" into singularities around which ECD charges are centered (appendix of [1]). This is a peculiarity not shared by the (integrable and conserved) ECD e-m tensor or electric current, both similarly diverging.

In Section 2.3, next, we discuss a mechanism allegedly "fixing" the scale of all particles of the same specie to their common value, yet we shall argue in both contexts of particle physics and cosmology that we actually do observe also scaled versions of those particles.

When shifting to a different scale, the electric charge of a particle, whether elementary or composite, does not change by virtue of scale invariance of electric charge $\int d^3x\, j^0$. In contrast, the scaling dimension of the particle's magnetic dipole moment $\mu_i = \frac{1}{2} \int d^3x\, \epsilon_{ijk} x^j j^k$ is 1 and, hence, scale dependent. If, further, the particle is sufficiently isolated then, since the electromagnetic field in its neighborhood is dominated by its electric current, one can associate the global e-m tensor $P$ (7) in that neighborhood with the particle (or particles in the case of a composite), referring to it as $P^{(a)}$. The particle's self energy (or mass), $\int d^3x\, P^{(a)\,00}$, incorporating also the electromagnetic self-energy which is a finite quantity in ECD, therefore has scaling dimension $-1$, while its three dimensional angular momentum, $J_l = \int d^3x\, \epsilon_{ijk} x_j P^{(a)\,0k}$, is scale invariant. All these scaling dimensions become critical in Section 3, dealing with the consequences of scale transitions.

## 2.3. The Zero-Point Field and Broken Scale Covariance

Most physicists are not troubled by a theory's lack of scale covariance for the simple reason that nature does appear to single out privileged scales. All hydrogen atoms around, for example, have a common Bohr radius. However, all such atoms also have the same position (on cosmological scales), and this does not imply that translation covariance is not a symmetry of the equations

describing hydrogen atoms. It does mean, nonetheless, that, for scale covariance to be compatible with observations, some dynamical process must exists in the universe, causing all hydrogen atoms to "cluster in scale", similar to the way gravity induces spatial clustering.

Continuing the above analogy, for matter to cluster in space, its position must be able to change. Likewise, for it to cluster in scale, its scale must be able to change. As such, a "motion in scale" must still respect local e-m conservation, and given the scaling dimension, $-1$, of self-energy, a particle moving towards larger scales must deliver some of its e-m content to its retarded electromagnetic field, whereas to move towards smaller scales, it must gain e-m from its advanced field. In either case, this field affects all other particles in the universe (at the intersection of their world-lines with the light-cone of that segment of the particle's own world-line during which its scale changes). These reacting particles, in turn, affect their light-cone neighbors—future and past—and so on, resulting in a giant spacetime "maze of interconnections" ([7] p. 181), involving all particles in the universe at all times.

The total electromagnetic field at a point $x$ in empty spacetime (viz., $x$ belonging to the complement of the support of $\cup_a P^{(a)}$), containing both advanced and retarded components, will be referred to as the *zero-point field* (ZPF) at $x$, a name borrowed from Stochastic Electrodynamics (SED) [8] although it does not represent the very same entity (in particular, our ZPF is *not* some global homogeneous solution of Maxwell's equations). Any system for which the time-averaged integrated ZPF Poynting flux across a sphere containing it vanishes is said to be in energetic equilibrium with the ZPF (in [6], dealing with the change of a system on cosmological time scales, we prefer the name "scale equilibrium").

Next, we assume that the density of matter in the universe is sufficiently high so as to result in 'complete absorption' of any particle's self-field. By this we mean that the energy sum of all particles in the universe fluctuates around a constant value, with the role of the ZPF reducing to that of a 'middleman', exchanging e-m between different particles (most of which are in their ground energy-equilibrium state, hence the name ZPF) securing (total) e-m conservation. It will further be assumed that the marginal energy distribution of any single particle depends only on its specie (electron, helium atom etc.) and not (significantly) on its location relative to other particles.

The above premise of location independence is natural for two reasons. First, the ZPF is dominated by a particle's self-field in its vicinity. As far as its ECD solution is concerned, the role of other particles' self-fields is restricted to imposing on it a given average energy/scale. That they are able to do so, despite being negligible compared with the particle's self-field, is consistent with $\alpha$ in (10), which controls scale motion, depending (highly) nonlinearly on them. Second, in a universe which is homogeneous beyond a certain length scale, the average number of particles in a thin shell of radius $r$, centered at any given particle, is $\propto r^2$, compensating for the $r^{-2}$ drop of a radiation field. The tiny correction to a particle's self-field is therefore an attribute of the entire universe not diverging due to mutual destructive interference brought about by their previously discussed interconnectedness.

The local intensity of the ZPF, nonetheless, obviously does depend on that distribution of matter. As one gradually gets closer to some concentration of matter, the local statistical properties of the ZPF become increasingly more dependent on the specific form of the associated matter distribution. In [5], it was shown how such matter-induced modulations of the ZPF are incorporated into QM wave equations, constituting the mechanism by which a particle can "remotely sense" a distant object, such as the status of the slit not taken by it in a double slit experiment. In Section 4.1, we shall argue that those modulations in the ZPF further offer an appealing explanation for "dark-matter". In [6], we "close the loop", tying the very small with the very large. A preferred scale, such Compton's length, resulting from scaling symmetry breaking, is completely arbitrary in Minkowski's space but not so in a Friedman universe, where the ZPF is a source of cosmological curvature. With the loop closed, a radically different interpretation of astronomical data ensues.

### 3. ECD and Particle Physics

If asked about the nature of the atomic world, a chemist would reply roughly as follows: Matter is made out of heavy, positively charged nuclei with light, negatively charged electrons frenetically moving in between them, thereby countering the electrostatic repulsion between the nuclei (why the electrons do not radiate their energy and spiral towards a nucleus—(s)he neither knows nor cares). Schröedinger's equation simply describes the time-averaged joint charge distribution of those constituents which, for stable molecules, should indeed be time independent.

On hearing the chemist's reply, a physicist would object that such a description cannot possibly be what is *really* happening. For when a molecule is ionized, the Schrödinger wave-function of even a single electron gets spread over a huge region, which is incompatible with a particle description of an electron. When, furthermore, two electrons are ejected in an ionization process, the chemist's picture makes even less sense.

In [5], it is shown that the chemist's simple and intuitive picture is consistent with everything physicists know about Schröedinger's equation and atomic physics alike, including quintessentially quantum mechanical phenomena such as those involving entanglement, spin-$\frac{1}{2}$, and even photons. Moreover, the chemist's disregard for radiation losses is fully warranted, while the physicist's problem with the spread of the wave-function stems from a confusion between time and ensemble averages: the charge of an electron is, indeed, confined to a tiny region. The multi-particle wave-function describes the joint charge distribution of an ensemble of *different* systems, but in (quasi-) equilibrium scenarios, and there only, such as those often described in chemistry, the ensemble averages can be replaced by time averages of a *single* system, much like in statistical mechanics of ergodic systems.

There is not a single experimental evidence, we argue in this section, suggesting that the chemist's picture should not apply to the subatomic domain and particle physics in general, and that additional interaction modes beyond the electromagnetic one at all exist on those smaller scales. In other words, the ontology of particle and nuclear physics could still be that of classical electrodynamics, provided, of course, that classical electrodynamics is given a consistent meaning, which is what ECD is all about.

So, why do we not apply the chemist's methods also to atomic nuclei and particle physics in general? After all, it is remarkably efficient compared with the standard model of particle physics—which, one must add, is almost useless when it comes to nuclear physics: a single multi-particle Schrödinger's equation with three tunable parameters capable of describing the morphology, strength, and other physical and chemical properties of millions of different complex molecules, compared with the standard model, whose mathematical structure is astronomically more complicated (and ugly, some would say) and with a comparatively lame output—resonance energies, lifetimes, and cross sections.

The reason for the failure of the chemist's description on subatomic scales, we argue, is not that a different ontology characterizes the subatomic domain but, rather, that Schrödinger's equation, and QM wave equations in general, are applicable only in those cases in which the effects of self electromagnetic interaction can be "absorbed" into the parameters of the equation, and it just happens that this is the case at the atomic scale but not on the much smaller scale involved in nuclear/particle physics. More precisely, we showed in [5] that, for QM wave equations to properly incorporate self-interaction, their associated charge distribution must be much wider than the width of the (extended) particle they describe. It should therefore not come as a surprise that the constituents of a proton, for example, densely packed into a tiny volume compared with that of an atom, do not necessarily satisfy this condition (see bellow).

The collapse of Schrödinger's description at subatomic scales is so colossal that one has to basically work out from scratch a new statistical theory, treating self electromagnetic interaction non perturbatively (unlike in QED). If ECD is indeed a valid description of the subatomic ontology, then settling for the lame phenomenology provided by the standard model would be tantamount to keep using Ptolmaic epicycles in contemporary astrophysics—a fairly accurate description, but extremely limited in its scope. Regrettably, this is easier said than done.

*3.1. A Tentative Ontology Based Solely On ECD*

In light of the above introduction, it is stressed again that the following is *not* a proposed substitute to the standard model of particle physics but, rather, a proposed ontology, possibly underlying the statistical predictions of the standard model.

3.1.1. Charged Leptons

Electrons and their antiparticles, positrons, are the only stable elementary particles in our model, represented by a single particle ECD solution (Section 3.2 of [1]). Conversely, it is assumed that no charged, isolated, stable single particle ECD solution exists other than that representing an electron.

The spin of the electron does not necessarily involve spin-$\frac{1}{2}$ ECD (Appendix E of [1]) and may be due to an internal current in a scalar ECD solution. Which of the two will be decided by explicit calculations.

As the electromagnetic field in an electron's immediate neighborhood is dominated by its self-field, the ZPF (the part of the electromagnetic due to all particles but the isolated electron) is ignored in a first approximation, thereby restoring ECD's scale covariance, and our electronic ECD solution is defined only up to a scale transformation (13). It is conjectured that the ECD solutions representing the $\mu$ and $\tau$ leptons are just scaled versions of the electroinic solution, with their respective Compton lengths, $\hbar/(mc)$ being the characteristic size of their associated distributions. As explained in [5], an extended electron model not only does not conflict with experiment but also can remove known inconsistencies from Dirac's equation.

Clear support for the above scaling conjecture comes from a few simple observations which, in the standard model, appear simply as axioms. Recalling from Section 2.2 the scaling dimensions of the electric charge (0), angular momentum (0), magnetic dipole moment (1), and self-energy ($-1$), the fact that all charged leptons share a common charge and intrinsic angular momentum but differ on their magnetic moment by a factor which is inversely proportional to their mass receives a simple explanation.

The role of the neglected ZPF in our model is to give each of the scaled solutions an effective life-time (and tiny corrections to their $g = 2$ gyromagnetic constant), and only three apparently make it to an observable level. The fact that different scaled versions have different lifetimes is clearly a bias of the ZPF which is *not* expected to be scale-invariant, given that every other aspect of our universe is not scale invariant either.

3.1.2. Hadrons

Hadrons are speculated to be composite rather than elementary ECD particles. The notion of "composite" in ECD, however, has a very different meaning from its standard-model counterpart, where it stands for a bound aggregate of elementary particles, such as quarks, each with definite autonomous attributes. Instead, it represents a multi-particle bound solution of the ECD equations. The distinction is critical because of the highly nonlinear nature of ECD. When elementary ECD particles cluster to form a composite, possibly overlapping, that nonlinearity renders their previous attributes completely irrelevant, and a genuinely new type of particle is formed.

There is, however, one exception to the above identity loss on the part of elementary ECD particles: electric charge, which is the only conserved quantity associated with individual particles. It follows that, if each constituent of a composite is *somewhere* along its (extended) world-line a free electron/positron, then *t*he common quantization of the electric charge in all particles is trivially explained. The equality in magnitude between the electron's and the proton's electric charges, which is verified to the utmost precision by the overall electric neutrality of ordinary matter, appears in the standard model as an ad hoc postulate involving electrons' and quarks' charges and must trouble any physicist seeking simplicity in the laws of nature.

In somewhat greater detail, we propose the following model of hadrons. Baryons are metastable composites made of three ECD charges. In addition, they can also have between one and three weakly bound charges, all of opposite charge. An archetype discussed below (Section 3.1.3) is the neutron, whose "core" is a proton. Neutral mesons are made of two, oppositely charged ECD charges, and charged mesons have an additional weakly bound charge, e.g., $\pi^{\pm}$ whose core is a $\pi^0$. Note that, the addition of a weakly bound charge can either increase or decrease the core's self-energy/mass, a phenomenon familiar from nuclear fusion.

Now that the relation between elementary and composite ECD particles is established, we can see in more detail why QM wave equations cannot describe hadrons. Concretely, an ECD proton is supposed to comprise two positively charged ECD particles and a negative one, all fitting into a ball of radius $R \sim 10^{-15}$ m. Given that the electron's size is about three orders of magnitude larger than $R$ and the scaling dimension $(-1)$ of mass (self-energy), we need to scale up the mass of an electron by at least four orders of magnitudes for it to freely fit into a ball of radius $R$ (and hence be amenable to Schrödinger's equation) giving a proton mass which is at least an order of magnitude too high even if we neglect the electromagnetic binding energy. This means that each ECD constituent of a proton must have a size comparable with $R$, with significant overlaps between different constituents.

Finally, it is conceivable that some "baryon resonances" are charged, single particle ECD solutions not related to the electronic solution via a scaling transformation but constrained by the electron's charge. At present, we cannot anticipate whether these are stationary ECD solutions—not necessarily time independent but with a regular, periodic dependence on time—or chaotic ECD solutions, of the type representing atoms and molecules.

### 3.1.3. Nuclei

Fundamentally speaking, atomic nuclei are just large ECD composites. Practically speaking, this is not a particularly useful insight, so we shall resort to an intermediate level of abstraction, involving the proton, chosen both for its absolute stability and because of the role its mass plays in quantizing—albeit only approximately—the atomic masses of all elements, their isotopes included.

The simplest nontrivial nucleus is that of a deuterium atom, and its ECD representation is not qualitatively different from that of a $H_2^+$ ion: two protons, plus a negative light particle, frenetically moving (mostly) in between them, thereby countering their mutual Coulomb repulsion—a so-called covalent bond.

The obvious difference between the above two systems is their size, which is about four orders of magnitudes larger for $H_2^+$. This is apparently the reason why, historically, the appealing (and extremely successful!) picture of a covalent bond was rejected from the outset in early attempts to model atomic nuclei. Nonetheless, by our previous remarks concerning hadrons, it is not that the qualitative picture of an electromagnetic covalent bond must fail at such small distances but, rather, that, at this smaller scale, Schrödinger's equation fails to consistently describe its statistical properties. Moreover, in this regime, the binding negative particle cannot possibly remain an electron with a size larger than that of the deuterium nucleus by three orders of magnitude. Instead, it is some negative ECD particle contributing little to the overall energy of the system, and only when it escapes the nucleus alone (e.g., in $\beta^-$ decay) does it eventually assume one of the stable single-particle ECD states, which are charged leptons. When a proton is further released in a nuclear decay, the two could bind to form a metastable neutron and, again, (mainly) the negative particle "morphs" into a new identity imposed by the different host.

This picture of a neutron—that of a negative particle weakly bound to a proton—is consistent with the neutron's subsequent decay into a proton, an electron, a (anti-)neutrino, and possibly a photon, the latter two—we argue below—being just manifestations of absorption of electromagnetic radiation created by the jolting of the electron.

The covalently bound-proton model of nuclei further explains the existence of a so-called "belt/band of stability" in the protons vs. neutrons chart of radioisotopes (which, in our interpretation,

is a protons-minus-negative-charges vs. negative-charges chart). The stability of a nucleus with a given number of protons clearly depends on the number of negative charges covalently binding them. Too little of them, the Coulomb interaction may favor splitting the nucleus. Adding more of them, however, does not increase its binding energy indefinitely. Beyond a certain number, attained at the belt-of-stability, any added binding charge must come at the expense of an existing one (roughly speaking, two such charges cannot both reside in between two protons because of their mutual repulsion). An excess of such negative binding charges allegedly leads to $\beta^-$ decays. A deficit, in contrast, could have more diverse manifestations. Nuclear fission was already mentioned; an electron captured from the atom clearly gets the nucleus closer to the belt but also the creation of a charged pair inside the nucleus, followed by the release of the positive particle which, outside the nucleus, morphs into a positron ($\beta^+$ decay). Finally, the large ($p \gtrsim 10$) atomic number part of the belt can be nicely fitted by a curve derived from two reasonable assumptions only: (1) The number of negative charges is proportional to that of the protons, minus a term proportional to the surface area of the nucleus (protons on the surface have fewer neighbors), and (2) the volume of a nucleus is proportional to the number of its protons (which is not its atomic number in our model).

### 3.1.4. (The Illusion of) Photons and Neutrinos

Photon- and neutrino-related phenomena embody, perhaps, the most drastic consequence entailed by the inclusion of advanced fields in ECD. To set the stage for their appearance, let us first review the standard classical model of radiation absorption which must obviously be revised.

Suppose, then, that a system decays to a lower energy state, releasing some of its energy (and possibly also linear and angular momentum) content in electromagnetic form. The retarded electromagnetic pulse carrying this energy subsequently interacts with other systems in which the response entails generation of a secondary retarded field, superposing destructively with the original at large distances, thereby attenuating the pulse's Poynting flux in its original direction. If the response of an absorbing system does not generate a Poynting flux in directions other than that of the original pulse, the process is classified as absorption. Otherwise, it incorporates, to some degree, also scattering. Ultimately, possibly following many scattering process, when the pulse is fully absorbed by matter, its e-m gets reconverted to "mechanical form", now appearing in the absorbing systems. This complete reconversion means that the (retarded) Poynting flux on a sufficiently large sphere containing the decaying system and the absorbing medium must vanish.

Two modifications to the above description are mandatory when advanced solutions are included. First, neither retarded nor advanced fields on that large sphere can ever vanish due to the existence of the ZPF. However, for the e-m content of the decaying system to remain inside the sphere, it suffices that the time-integral, over the Poynting-flux integral across it, should vanish. This, in turn, is just the definition of the system comprising the interior of the sphere being in equilibrium with the ZPF, meaning that the absorption of radiation only amounts to a transition of matter inside the sphere, between distinct equi-energetic equilibrium states. Second, ECD systems could also "undecay"—get energetically exited. A decaying system in our universe is characterized by a sudden imbalance between its retarded and advanced self-fields, favoring the former. In exited systems, that imbalance favors the advanced field. In this case, as well, we postulate that no time-integrated Poynting flux imbalance appears on a sufficiently large sphere containing both the exited system and the system(s) where an energy deficit must appear by e-m conservation.

In *macroscopic* systems, the imbalance between advanced and retarded (Poynting) powers constitutes only a small fraction of the combined powers. More formally, we decompose the current associated with a macroscopic system into an average + fluctuations:

$$j = J + \tilde{j}, \quad J := \langle j \rangle, \quad \langle \tilde{j} \rangle = 0, \tag{14}$$

where $\langle \cdot \rangle$ stands for a convolution with some normalized, Lorenz invariant kernel, whose support is much larger than typical atomic size yet much smaller than the scale of variation of $J$. The full potential (10) can then be written:

$$A = K_{\text{ret}} * J + \alpha_{\text{adv}} \left( K_{\text{adv}} - K_{\text{ret}} \right) * J + \left( \alpha_{\text{ret}} K_{\text{ret}} + \alpha_{\text{adv}} K_{\text{adv}} \right) * \tilde{J} \tag{15}$$

with $*$ the convolution in (10) and obvious indices omitted. The first term in (15) is conventionally referred to as the "retarded potential of a source". The second term is a homogeneous solution of Maxwell's Equation (2) and, as such, does not affect the e-m balance of a system on macroscopic time-scales. This term can therefore be consistently ignored if the corresponding term in the receiving antennas is. Nonetheless, it *does have* observational consequences, e.g., in the Bohm–Aharonov effect. There, the "classical" magnetic field outside an infinite solenoid, due to the first term in (15), vanishes, but since $\alpha_{\text{adv}}$ is not identically constant, the second term does not. QM then allegedly describes the statistical consequences of this meager field [5]. The final term in (15) corresponds to the zero-point motion of charges. As we shall see in Section 4.1.1, consistency with observations mandates that it should dominate (15).

If one excludes advanced fields, as historically was the case in CED, then in an excitation scenario, conjectured to apply, e.g., in the ionization of an atom, an electron is suddenly seen ejecting at high speed with no apparent energy source to facilitate such a process. This had led Einstein to hypothesize a neutral massless particle whose collision with the electron resulted in ionization—a hypothesis which agonized him for the rest of his life.

The symmetry between "photon production" by a system, viz. transitions favouring the retarded self-field, and "photon absorption" (advanced field favoured), which is assumed to hold at the microscopic scale, is broken at the macroscopic scale by the arrow-of-time. Photons can be produced by decaying microscopic systems, such as a molecule, but also by a (macroscopic) burning candle, or in Bremsstrahlung, among others. Absorption of photons, in contrast, involves the excitation of microscopic systems only. This asymmetry creeps into the quantum mechanical description of radiation absorption, in which the absorbed (retarded) radiation enters as a classical field into the wave equation. A typical example is the ionization/excitation of a molecule by a weak external electromagnetic pulse, assumed to be generated by some macroscopic source, such as a laser. A standard result of time-dependent perturbation theory combined with the dipole (long wave-length) approximation and the "ensemble interpretation" of the wave-function (see Section 4 in [5]) imply that the molecule acts as a spectrum analyser for the pulse, with the number of its transitions between states of energy gap $\Delta E$ being proportional to the spectral density of the pulse at frequency $\Delta E / \hbar$. This result explicitly demonstrates the vanity of expressions such as a "blue photon".

The external pulse, of course, is not limited to the relatively low frequencies generated during atomic transitions. However, as the frequency is increased towards the $\gamma$ part of the electromagnetic spectrum, there are, in general, fewer systems for which the transitions involve the generation of such high-frequency secondary retarded waves (needed for absorption of radiation), increasingly necessitating atomic nuclei to this end. This fact, according to our model, is the reason for the greater penetration power of high-frequency pulses rather than the "greater energy of high-frequency photons". Similarly, their greater destructive power is explained by the fact that, in order for the absorbing system to produce a high-frequency secondary pulse, its electric current during the transition must, likewise, have high-frequency components, implying a more violent response on the part of the absorbing system. (Note that we cannot naively extrapolate the previous results of QM wave equations applied to atomic transitions to arbitrarily high frequencies, as by our opening remarks for this section, QM wave equations no longer apply to atomic nuclei, hence the need for heuristic arguments.)

It is, however, only when photons are "created" in the decay of a *microscopic* system that the consequences of including advanced fields have their most dramatic effect. According to QM, assumed to correctly capture statistical aspects of ECD solutions, the equilibrium states of bound matter systems are extremely rare. Rather than having the continuum of allowed time-averaged energy

and angular momentum that bound classical systems have, theirs assume discrete values. If we now combine (a) complete absorption, (b) e-m/angular momentum conservation, and (c) severe constraints on ECD equilibrium solutions, we get that the e-m lost in the decay of the microscopic system must not always appear continuously spread over the interior of the absorbing sphere (so-called soft photon absorption). In some cases, that entire energy/angular momentum deficit of the decaying microscopic system, temporarily stored in the ZPF, reappears in compact, possibly remote systems, e.g., single atoms. Moreover, systems directly exposed to the pulse released in the decay of the microscopic system are obviously more likely to be included in those absorbing "chosen ones" (consistent with the results of QM, treating the pulse classically); hence, the event associated with the emission of photons would lie on the past light cone of the event interpreted as a subsequent absorption thereof.

　　　Our conjectured model of photons-related phenomena can, of course, work only through the "intimate collaboration" of all the systems inside the sphere. This collaboration is not intermittent, restricted to epochs of photons "emission and absorption", but rather a permanent one. A local collection of interacting particles, such as the gas molecules filling a particle detector or even an entire galaxy (see Section 4.1 below) must necessarily exhibit such a collaboration for it to remain in equilibrium with the ZPF. This collaboration, however, must not be understood in the sense of information-exchange, with signals running forward and backwards in time (whatever that means). In the block-universe, one has to stop thinking in dynamical terms, treating an entire process as a single "space-time structure", constrained by the ECD equations—the basic tenets included in them—and by QM which covers statistical aspects of ECD solutions.

　　　Neutrinos. Neutrinos' alleged "generation", "absorption", and "scattering" (e.g., $n \rightarrow p + e^- + \bar{\nu}_e$, $\nu_x + d \rightarrow p + p + e^-$ and $\nu_x + e^- \rightarrow \nu_x + e^-$ resp.) all involve manifestly radiating systems—jolted charge(s)—and in this regard, they are very similar to energetic photons. Like photons, neutrinos seem to propagate at the speed of light, as the SN 1987A supernova clearly shows (unless "God is malicious") in conjunction with artificially produced neutrinos all traveling at light speed to within measurement error. The need for a distinct category (actually three of them) was born out of the necessity to salvage energy and angular momentum conservation in $\beta$ decay, as no "photons" were detected which could have done the job (and photons were already "assigned" an incompatible, integer angular momentum). However, ontologically, neutrino-related phenomena are indistinguishable from that of photons. The extreme "penetration depth" of neutrinos is explained by the same argument used above in the case of $\gamma$ photons: There are allegedly almost no systems for which excitation entails the generation of an electromagnetic field, destructively superposing with the incident field (generated in a process associated with the production of a neutrino). In the current case, however, the scarcity of such systems is not due to the required extreme frequencies but probably to unique, wide band wave forms.

## 4. ECD and Astrophysics

　　　The ZPF has an ultra low energy density compared with that of matter, and a practically vanishing Poynting vector. It is practically ignorable on everyday, macroscopic scale. In Section 3 and in [5], we speculated that when diving deep into the atomic and subatomic domains, the tiny masses of particles render the ZPF consequential to their behavior. In the current section we argue that also by moving in scale in the opposite direction, towards galactic and ultimately cosmological scales, the effects of the ZPF become manifest, amplified this time by the huge volumes involved, over which even a meager density can integrate to a large value.

　　　Analyzing ECD's consequences for astrophysics requires first that it be consistently fused with general relativity. As advocated in the introduction, this is done by expressing flat spacetime ECD

(Maxwell's equations included of course) in an arbitrary coordinate system via the use of a "metric" $g_{\mu\nu}$. These equations are supplemented by Einstein's field equations

$$\mathcal{R}_{\mu\nu}\left[g_{\mu\nu}\right] - \Lambda g_{\mu\nu} = 8\pi G \left( P_{\mu\nu} - \frac{1}{2} g_{\mu\nu} P^{\lambda}_{\;\lambda} \right), \tag{16}$$

with $P$ the generally covariant e-m tensor and $\mathcal{R}$ the expression for the Ricci tensor in terms of the metric, $g_{\mu\nu}$, and its derivatives: $\mathcal{R}_{\mu\nu}[g] := \partial_\rho \Gamma^\rho_{\;\nu\mu} - \partial_\nu \Gamma^\rho_{\;\rho\mu} + \Gamma^\rho_{\;\rho\lambda}\Gamma^\lambda_{\;\nu\mu} - \Gamma^\rho_{\;\nu\lambda}\Gamma^\lambda_{\;\rho\mu}$. Equation (16) corresponds to the most general choice of coefficients in (1) for which its l.h.s. is covariantly conserved (by virtue of the second Bianchi identity). This form is mandated by ECD, whose energy-momentum (e-m) tensor, $P_{\mu\nu}$, is by construction covariantly conserved, $\nabla^\mu P_{\mu\nu} = 0$. Note that this is *not* the argument given to this choice by Einstein [2].

The basic tenets (2)–(4) become their obvious generally covariant extensions. In particular, by the antisymmetry of $F$, Maxwell's equations simplify to

$$(a) \quad g^{-1/2}\partial_\nu \left( g^{1/2} F^{\nu\mu} \right) = j^\mu \qquad (b) \quad \partial_\lambda F^{\mu\nu} + \partial_\mu F^{\nu\lambda} + \partial_\nu F^{\lambda\mu} = 0, \tag{17}$$

while covariant e-m conservation reads

$$g^{-1/2}\partial_\mu \left( g^{1/2} P^{\mu\nu} \right) + \Gamma^\nu_{\;\mu\lambda} P^{\mu\lambda} = 0, \tag{18}$$

with $g := \left| \det g_{\mu\nu} \right|$ and $\Gamma$ as the Christoffel symbol. From (17a) and the antisymmetry of $F^{\mu\nu}$, one gets $\partial_\mu \left( \sqrt{g} j^\mu \right) = 0$ as a consistency condition, generalizing (3).

Using the same construction as in Appendix D of [1], one can then show that, if a coordinate system exists for which $g_{\mu\nu}$ slowly varies over the extent of the particle, then (18) implies that the path of the "center of the particle", $\gamma^\mu(s)$, (given a clear meaning there) is described by the geodesic equation:

$$\ddot{\gamma}^\mu = -\Gamma^\mu_{\;\alpha\beta}\dot{\gamma}^\alpha\dot{\gamma}^\beta, \tag{19}$$

with "dot" standing for the derivative with respect to any parametrization, $s$, of $\gamma$. From (19), we have that $\dot{\gamma}^2 = const$ along $\gamma$, from which follows $ds \propto \sqrt{d\gamma^2}$.

To define dark matter, we will also need the following decomposition. Let the exact (modulo a coordinate transformation) metric and ECD e-m tensor in our universe be given by $g_{\mu\nu}$ and $P_{\mu\nu}$ resp. Convolving $P_{\mu\nu}$ with a kernel wide enough for the result to be effectively constant on non cosmological length scales, we denote by $\tilde{P}_{\mu\nu}$ the resulting low-passed/smoothed functio, and let $\tilde{g}_{\mu\nu}$ be a solution of (16) for the low-passed source, viz.

$$\mathcal{R}_{\mu\nu}\left[\tilde{g}_{\mu\nu}\right] - \Lambda \tilde{g}_{\mu\nu} = 8\pi G \left( \tilde{P}_{\mu\nu} - \frac{1}{2}\tilde{g}_{\mu\nu}\tilde{g}^{\lambda\rho}\tilde{P}_{\rho\lambda} \right). \tag{20}$$

The "tilde tensors" $\tilde{g}$ and $\tilde{P}$ are therefore involved in dynamical changes on a cosmological time scale and are studied in [6] dealing with cosmology.

Next, changing coordinates to the locally comoving Minkowsikian frame, which implies $\tilde{g}_{\mu\nu} = \eta_{\mu\nu}$, and defining the fluctuations, $p_{\mu\nu}$ and $h_{\mu\nu}$ by

$$P_{\mu\nu} = \tilde{P}_{\mu\nu} + p_{\mu\nu}, \quad g_{\mu\nu} = \tilde{g}_{\mu\nu} + h_{\mu\nu}, \tag{21}$$

we substitute $P_{\mu\nu}$ and $g_{\mu\nu}$ from (21) in (16), assume $h_{\mu\nu} \ll \eta_{\mu\nu}$, expand $\mathcal{R}_{\mu\nu}[\eta_{\mu\nu} + h_{\mu\nu}]$ to the first order in $h_{\mu\nu}$, and get

$$-\partial_\lambda \partial^\lambda h_{\mu\nu} + \partial_\lambda \partial_\nu h_\mu{}^\lambda + \partial_\lambda \partial_\mu h_\nu{}^\lambda - \partial_\mu \partial_\nu h_\lambda{}^\lambda - \Lambda h_{\mu\nu} + 8\pi G \left( h_{\mu\nu} \tilde{P}_\lambda{}^\lambda - \eta_{\mu\nu} h^{\rho\lambda} \tilde{P}_{\rho\lambda} \right) = \quad (22)$$

$$16\pi G \left( p_{\mu\nu} - \frac{1}{2} \eta_{\mu\nu} p_\lambda{}^\lambda \right),$$

where, to first order in $h_{\mu\nu}$, raising of indices can be done with either $g_{\mu\nu}$ or $\eta_{\mu\nu}$ (note that (18) implies the conservation of $p_{\mu\nu}$ only to the zeroth order in $h_{\mu\nu}$, consistent with the two exchanging e-m). As in our treatment of Maxwell's equations, we assume that no sourceless gravitational waves propagate in the universe; hence, $h_{\mu\nu}$ is entirely due to $p_{\mu\nu}$ (or, $p_{\mu\nu} \equiv 0 \Rightarrow h_{\mu\nu} \equiv 0$, consistent with (20)). Since $p$ and $\tilde{P}$ are of the same order and $h$ is of the order of $Gp$, the last term on the l.h.s. of (22) is of order $h^2$ and hence neglected. Anticipating the results of [6], the $\Lambda$ term in (22) is likewise ignored in the current epoch of the universe for its relative smallness.

### 4.1. ECD and Dark Matter

Astronomical observations of galaxies and clusters thereof clearly show that, insofar as their e-m content is correctly estimated, Einstein's field equations cannot possibly apply to them. This could only mean that our understanding is grossly erred in either or both (1) gravitation and (2) particle physics (being the branch of physics dealing with $P^{\mu\nu}$). Much less direct, alleged difficulties with Einstein's equations + SM pertaining to even larger, cosmological scales and to the remote past are addressed in [6].

Modified gravity theories, such as MOND [9] and its relativistic extension TeVeS or the so-called $f(R)$ and scalar-tensor theories, have thus far failed to yield a dark-matter free account of all relevant observations. Modified gravity theories are further way more complicated (and ugly, most would argue) than Einstein's gravity, contain an infinite number of tunable parameters (e.g., a function, $f$, in the case of $f(R)$-gravity, or an interpolation function, $\mu$, in MOND), and have merely begun going through the stringent tests already passed by the original. With recent detections of gravitational waves, concurrently with the expected optical signal, a severe new constraint has been added, refuting most existing modified-gravity proposals.

On a more conceptual level we ask, along with some DM advocates, what is the point in astronomy and cosmology if they cannot be intimately linked with terrestrial-scale physics? Those two disciplines will forever remain limited to the *passive* collection of electromagnetic (and perhaps gravitational) waves from a single point-of-view. A deformation of terrestrial physics—which survived a plethora of *active* tests—experimentally inaccessible otherwise, is not really different from epicycles drawing and will always be achieved with enough of them—interpolation functions, additional fields and terms in the Lagrangian, etc.

The more pervasive view is that Einstein's gravity should be kept, salvaged by new forms of yet unknown, exotic "dark matter" ("transparent matter" would have been more appropriate), at most weakly interacting with normal matter. This conjecture might seem too flexible to ever be refuted. Indeed, given $g_{\mu\nu}$, estimated on dynamical grounds, gravitational lensing, etc., a compatible e-m tensor $P^*_{\mu\nu}$ is guaranteed to exist by Einstein's equations. Representing dark-matter by $P^{\text{Dark}} := P^* - P^{\text{Bar}}$ with $P^{\text{Bar}}$, the e-m tensor associated with directly observed baryonic matter, in conjunction with $P^{\text{Dark}}$ and $P^{\text{Bar}}$ at most weakly interacting (not via gravity) would always vindicate Einstein. Of course, $P^{\text{Dark}}$ should "balance" $P^{\text{Bar}}$ not only at some cosmological time but also throughout the entire history of the universe. However, given the speculative nature of that history, its highly indirect evidence, and the immunity granted to dark particles from being directly detected, dark matter does appear somewhat of a lazy solution.

The above, common criticism of the dark-matter conjecture ignores nonetheless a nontrivial observational fact: The energy density associated with $P^{\text{Dark}}$ is always found to be positive semi-definite,

as expected of matter. Why should that be so if GR is wrong? In other words, why do we not also observe systems with too much mass rather than too little? Our proposal that $P^{\text{Dark}}$ is just the e-m tensor associated with the ZPF explains this critical point. It further trivially explains the transparency of "dark matter" to electromagnetic radiation (by the linearity of Maxwell's equations). Finally, ordinary and (alleged) dark matter are observed clustered together, again, as if gravity does not discriminate between the two. This is explained in our model by the fact that the ZPF is just the sum of self-fields, each adjunct to an individual particle, declining with distance from it as must be the case for a radiation field. Regions rich in ordinary matter should therefore be also "ZPF rich".

The analysis which follows relies on Equation (22) for the fluctuations around the background. As in standard linearized gravity (See, e.g., [10] Section 10.1 but note the different sign conventions for the metric) a subset of solutions to (22) (with the last two term on its l.h.s. omitted) relevant to our case satisfies the simpler equation:

$$ -\Box h_{\mu\nu} = 16\pi G \left( p_{\nu\mu} - \frac{1}{2}\eta_{\nu\mu} p_{\rho\sigma} \eta^{\sigma\rho} \right) . \tag{23} $$

As $p$ still contains the fluctuations in the ZPF and the internals of atoms and molecules, both irrelevant to the dynamics of galaxies, we utilize the linearity of (23) and "low-pass" it, viz. convolve it with a space-time kernel much wider than typical atomic size/time. Retaining the symbol for the low-passed $p$, the resulting r.h.s. should be separately treated for matter and radiation dominated regions. Starting with the former, the following standard assumption is made regarding nonrelativistic matter: $p_{ij} = p_{i0} = 0$. Only diagonal elements of $h_{\mu\nu}$ are therefore nonvanishing, the time-derivatives part of the l.h.s. of (23) are obviously negligible for a slowly varying $p$, and Poisson's equation for the Newtonian gravitation potential, $\Phi$, follows by defining $\Phi := h_{00}/2$:

$$ \nabla^2\Phi = 4\pi G p_{00} , \tag{24} $$

implying $h_{ij} = 2\delta_{ij}\Phi$.

In regions void of matter, where $\sum_a T^{(a)} = 0$ and the ZPF dominates the r.h.s. of (23), the tracelessness of the canonical tensor $\Theta$ implies that the r.h.s. of (24) becomes $8\pi G p_{00}$, viz. twice the value expected from naive mass–energy conversion. Furthermore, unlike in the case of (nonrelativistic) matter, we cannot simply neglect $p_{ij}$ and $p_{i0}$, sourcing the corresponding components of $h$. Nevertheless, for nonrelativistic motion, which is the case in most of what follows, the geodesic Equation (19) is sensitive only to $h_{00}$—hence only to $p_{00}$—Reducing to Newton's equation:

$$ \ddot{\gamma} = -\boldsymbol{\nabla}\Phi(\gamma) , \tag{25} $$

with "dot" being the derivative with respect to $x^0$.

### 4.1.1. Outline of a ZPF-Based Model of Dark-Matter

No attempt is made in this short section to fully cover the astronomical observations concerning dark matter which have been occupying telescopes around the globe for several decades. Instead, we shall demonstrate that any reasonable ZPF-based model of dark matter qualitatively captures the more universal aspects of this huge body of knowledge. It is crucial to understand that a full-fledged model cannot be derived from ECD alone for exactly the same reasons QM cannot. Both should be viewed as complimentary statistical theories to ECD, on equal footings with the latter. As with QM, simplicity criteria can be *postulated* such that the model becomes the simplest, perhaps unique such compatible statistical theory, but the postulates themselves would obviously not be derivable from ECD.

According to our proposal, rather than inventing new forms of matter to explain the apparent deficit on the r.h.s. of (24), one has to take into account the effect which ordinary matter has on its

surrounding ZPF. Consequently our missing "transparent component" of $p_{\mu\nu}$ has several nonvanishing components besides $p_{00}$, clearly distinguishing it from ordinary cold dark matter—see Section 4.1.4.

The transparent $p_{\mu\nu}$ will be estimated from the paths, $x^{(a)}(t)$ of all relevant particles, labeled by $a$. Only their dipole fields will be used to represent their radiation (the Coulomb part, by our previous remarks, appears in the e-m of matter), but this is just to ease the presentation, with higher order multipoles adding nothing essentially new to the analysis. In this approximation, we have

$$\left. B^{(a)}_{\substack{\text{ret}\\\text{adv}}}(t,\boldsymbol{x}) = \frac{\boldsymbol{n}^{(a)} \times \ddot{\boldsymbol{p}}^{(a)}_{\substack{\text{ret}\\\text{adv}}}}{\left|\boldsymbol{x} - \boldsymbol{x}^{(a)}\right|}\right|_{t_{\substack{\text{ret}\\\text{adv}}}}, \qquad \boldsymbol{E}^{(a)} = \boldsymbol{B}^{(a)} \times \boldsymbol{n}^{(a)}, \tag{26}$$

and (26) is to be evaluated at the retarded/advanced time, defined by solutions to

$$t_{\substack{\text{ret}\\\text{adv}}} - t = \mp \left|\boldsymbol{x} - \boldsymbol{x}^{(a)}\left(t_{\substack{\text{ret}\\\text{adv}}}\right)\right|$$

Above, $\boldsymbol{B}^{(a)}$ and $\boldsymbol{E}^{(a)}$ are the associated magnetic and electric fields; $\boldsymbol{x}^{(a)}$ is its c.o.m.; and $\boldsymbol{n}^{(a)} = \left(\boldsymbol{x} - \boldsymbol{x}^{(a)}\right)/\left|\boldsymbol{x} - \boldsymbol{x}^{(a)}\right|$ is a unit vector pointing from it at the point of interest, $\boldsymbol{x}$. The particle's effective dipole moment is

$$\boldsymbol{p}^{(a)}_{\substack{\text{ret}\\\text{adv}}}(t') = \int \mathrm{d}^3 y \, \alpha_{\substack{\text{ret}\\\text{adv}}}(t',\boldsymbol{y})\boldsymbol{y}\varrho^{(a)}(t',\boldsymbol{y}), \tag{27}$$

with $\varrho^{(a)}$ as its charge density and "dot" standing for a time derivative.

The derived low-passed (see the following (23)) electromagnetic energy density,

$$\Theta_{00}(\boldsymbol{x},t) = \frac{1}{2}\left(E^2_{\text{total}} + B^2_{\text{total}}\right), \tag{28}$$

involves both a double summation (26) over the particle labels and a separate count for their advanced and retarded contributions. It is clearly still time-dependent, but we shall only consider systems for which this time dependence can reasonably be assumed to be negligibly slow. As the time-averaged masses of the particles are assumed to be constant, the retarded and advanced contributions are equally weighted, reflecting $\langle\alpha_{\text{ret}}\rangle = \langle\alpha_{\text{adv}}\rangle = 1/2$ in (10). Strictly speaking, equally weighting retarded and advanced contributions is wrong, as the decomposition (15) implies. However, insofar as the last term in (15)—that associated with the ZPF—when integrated over scales of an entire galaxy is on the order of the galaxy's baryonic mass, the first term—that associated with macroscopic retarded radiation—and hence also the second term, can both be easily shown to be negligibly small in comparison.

The remaining components of $\Theta_{\mu\nu}$ will only interest us in Section 4.1.4, dealing with gravitational-lensing tests of dark matter. Qualitatively, they are as follows. Far from a dipole, the local radiation field is well approximated by a sum of plane waves, each of the form $A^{\mu}(x) = \epsilon^{\mu}f(k\cdot x)$, $k^2 = k\cdot\epsilon = 0$, $\epsilon^2 = -1$ for some $f$, with an associated canonical tensor $\Theta^{\mu\nu} \propto k^{\mu}k^{\nu}(f')^2$. Advanced and retarded contributions of each dipole to the local ZPF have opposite signs for their $k_i$s but the same for their $k_0$s. Since the ZPF receives nearly balanced contributions from retarded and advanced fields (Figure 2), the Poynting vector $\Theta_{0i}$ is negligible compared with $\Theta_{00}$ and $\Theta_{ij}$.

Had all dipoles been independently radiating, only "diagonal terms" in (28), viz. $a = b$ in (26), would have contributed, resulting in a trivial sum of "$r^{-2}$ halos" centered at every dipole. Not only would that render the "Friedman DC", $\tilde{P}^{00}$, infinite, but also it would further contradict the high degree of interparticle connections, discussed in Section 2.3 and in the context of our classical photon model (Section 3.1.4). Such interconnectedness mandates a certain degree of statistical dependence between any two dipoles at the intersection of their world-line with the other's light-cone. A premise of any consistent model must therefore be that this statistical dependence is destructive, viz. results in a (statistically) negative contribution to (28).

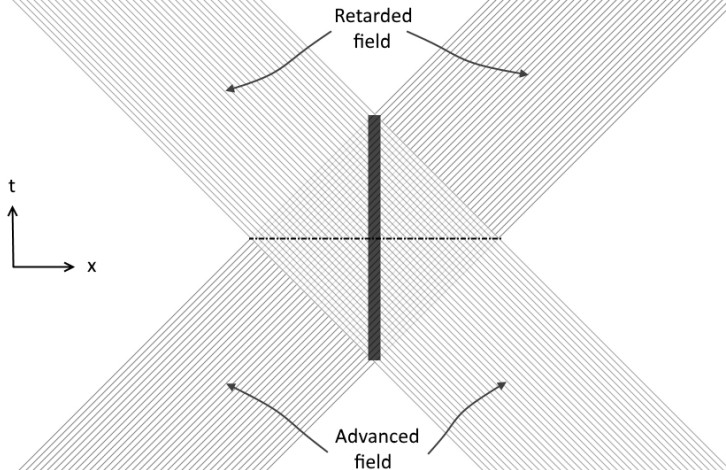

**Figure 2.** The thick vertical line represents the spatially extended world line of dense nebula $\mathcal{N}$ (existing for a finite time for illustrative purpose only), and the dashed horizontal line represents the constant-time three-surface to which our analysis of the outer halo applies (Equations (30) through (34)).

The destructive interference we refer to above is similar to the classical process of absorption discussed in Section 3.1.4, dealing with photons, but with one critical difference: There, the destructive interference between the incident retarded field and the secondary retarded field, generated by the absorbing system, entails the excitation of that system in order to respect energy-momentum (e-m) conservation. In the current case, the incident retarded field superposes destructively also with the advanced field of the absorbing system (see Figure 3). This destructive interference guaranties that the Poynting flux across a sphere, $S$, containing the absorbing system (or, as it should more appropriately be called in this case: the reacting system), vanishes, respecting its equilibrium with the ZPF. Reversing the roles of advanced and retarded fields, the advanced field of system b is likewise absorbed by system a. At the level of equilibrium with the ZPF, the arrow-of-time is inconsequential.

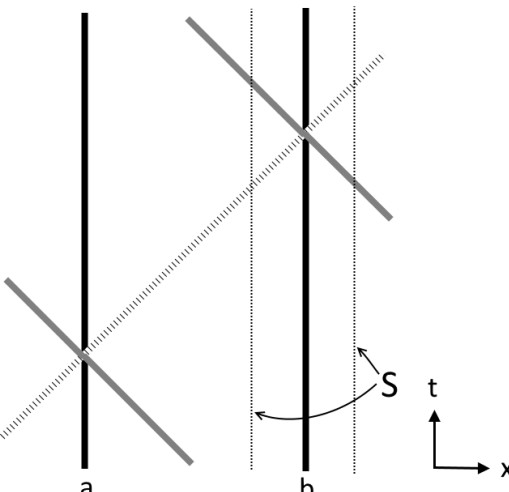

**Figure 3.** Mutual absorption between two particles in equilibrium with the zero-point field (ZPF): the dashed ray represents the locus of destructive interference. Note that, in $3 + 1$ spacetime, radiation fields decay with distance (26), implying that the degree of interference is minimal near each dipole, transversely extending beyond the ray, and its overall effect decreases with increasing interparticle separation.

Which dipole pairs are candidates for such destructive correlations? The interconnectedness of all dipoles notwithstanding, any realistic model can safely neglect all pairs, $(a, b)$, other than those for which $x^{(a)}$, $x^{(b)}$ and $x$ are approximately collinear. Otherwise, their correlation would necessitate at

least a third dipole, $c$, whose world-line intersects the intersection of $a$'s and $b$'s light-cones—a circle at a constant-time slice. Not only is that improbable (zero probability for point dipoles) but it would also be "second order" in a typical inter dipole correlation strength. Now, the probability of a random $x$ being collinear with any pair $(x^{(b)}, x^{(b)})$ is also zero. However, for any pair, there exists an entire line, $\{x\} : x \times (x^{(a)} - x^{(b)}) = 0$, satisfying the collinearity condition. It is along such lines, any model must assume, that destructive interference occurs. Now, for $n$ dipoles there are $O(n^2)$ pairs (whose negative contributions to the energy density (28) are clearly interdependent, decreasing with $n$ on average if only to guarantee (28)'s positivity). In conjunction with the diminishing distance-dependence of that interference (as explained in the caption of Figure 3), it follows that, for a given volume, $V$, hosting the dipoles, the average interference increases with increasing $n$ and decreasing $V$, or equivalently

$$\textit{significant interference occurs in high dipole-density regions} \tag{29}$$

Consider next a highly inhomogeneous distribution of dipoles comprising a relatively dense "nebula", $\mathcal{N}$, embedded in a tenuous background, $\mathcal{B}$. Assuming that the contribution to the energy density (28) coming from background pairs $a, b \in \mathcal{B}$ is effectively $x$-independent in the (large) neighborhood of $\mathcal{N}$ due to large numbers and sparsity of $\mathcal{B}$, and that contributions of mixed pairs, $a \in \mathcal{B}, b \in \mathcal{N}$ are negligible in that neighborhood (by the remarks made in the caption of Figure 3), one is left with nebula pairs of only $a, b \in \mathcal{N}$, responsible for the ZPF's contribution to $p_{00}$. For a nebula in which the center coincides with the origin and for $|x| \gg \max\{|x^{(a)}|\}$, viz. away from the nebula, we can Taylor expand (26) in powers of $x^{(a)}/|x|$, resulting in a (typically non-spherically symmetric) transparent "halo" (28):

$$p_{00}^{\text{halo}}(x) \sim \frac{f(\hat{x})}{|x|^2}, \tag{30}$$

$$f(\hat{x}) := \sum_{a,b} \sum_{\alpha,\beta \in \{\text{ret,adv}\}} \ddot{p}_\alpha^{(a)}(t_\alpha) \cdot \ddot{p}_\beta^{(b)}(t_\beta) - \ddot{p}_\alpha^{(a)}(t_\alpha) \cdot \hat{x} \, \ddot{p}_\beta^{(b)}(t_\beta) \cdot \hat{x} \geqslant 0. \tag{31}$$

For sufficiently large $|x|$, the assumptions underpinning (30) collapse and $p^{\text{halo}}$ merges seamlessly with the uniform coarse grained background $\tilde{P}_{\text{ZPF}}$, rendering the energy of the halo finite. This facilitates the calculation of its generated gravitational potential, $\Phi$ (24), using the standard Green's function of the Laplacian (which assumes a vanishing potential at infinity):

$$\Phi(x) \sim -\frac{G}{4\pi} \int_{S^2} \mathrm{d}\Omega f(\Omega) \int_{r_0}^{r_1} \mathrm{d}r \frac{1}{\sqrt{|x|^2 + r^2 - 2|x|r\cos\theta}} \tag{32}$$

with $\theta$ the angle between $\hat{x}$ and the direction of the spatial angle $\Omega$. The upper integration limit, $r_1$, guaranteeing the finiteness of the integral, is a convenient substitute for the actual, integrable large $|x|$ tail of the halo. The result of (32) is

$$\Phi(x) \sim -\frac{G}{4\pi} \int \mathrm{d}\Omega f(\Omega) \ln\left(2\sqrt{|x|^2 + r^2 - 2|x|r\cos\theta} + 2r - 2|x|\cos\theta\right) \Big|_{r=r_0}^{r=r_1}. \tag{33}$$

The gradient of $\Phi(x)$ at $r_0 \ll |x| \ll r_1$ is entirely dominated by the lower integration limit, and is further independent of $r_0$. It has a radial component

$$\nabla\Phi(x) \cdot \hat{x} \sim \frac{GC}{|x|}, \quad C := \frac{1}{4\pi} \int \mathrm{d}\Omega f(\Omega) > 0, \tag{34}$$

which is just the force field generated by a standard (spherically symmetric) isothermal halo and a transverse component, viz. orthogonal to $\hat{x}$ (unless the ZPF halo (30) is spherically symmetric) similarly decaying as $|x|^{-1}$. For the important case of a disc nebula, the transverse component vanishes in the disc's plane (by simple symmetry arguments).

The constant $C$ above, being an average over $\mathcal{N}$'s "transmittance", $f(\Omega)$, in all directions, is some global attribute of the nebula, already depending on more specific details of a model. However, in any reasonable model, we would have the following:

(a) The constant $C$ increases with increasing average strength of dipoles comprising $\mathcal{N}$. Reviewing Section 3 in this regard, it is reasonable to anticipate that the main source of the halo (30) is radiation linked with the bulk acceleration of electrons. Indeed, for a particle to have a strong radiation field, it must have a large/rapidly varying effective dipole (27). Freely moving particles, by their minute size and conserved charge, have tiny multipoles (in their c.o.m. frame), while their scale equilibrium entails that the fluctuations of their $\alpha's$ in (10) around their mean, $\langle \alpha_{\text{ret}} \rangle = \langle \alpha_{\text{adv}} \rangle = 1/2$, are relatively small. This implies that significant bulk acceleration is a necessary condition, hence the dominance of electrons, which are by far the lightest among the stable particles.

Rapid bulk acceleration occurs whenever the free motion of a charge is perturbed by a rapidly varying potential. This clearly is the case for ECD electrons bound in atoms and molecules, but any sufficiently dense environment also guarantees such conditions. As most baryons in the current epoch are in the form of tenuous plasma, in which electrons are freely moving for many years between collisions, major ZPF sources can only be neutral galactic gas and stars (the latter, despite being plasma dominated, are some thirty orders of magnitude denser than the intergalactic medium). This conclusion is further consistent with the results of [6], where a lower bound, $\epsilon \gtrsim 2$, was calculated for the ratio between the current average energy density of baryonic matter to that of the ZPF. Had the intergalactic medium generated ZPF in a proportion to its mass, on par with that of galaxies, $\epsilon$ would have had to be much smaller.

Correlating with the number of electrons in a nebula, the constant $C$ in (34) also correlates with the nebula's mass, the reason being the dominance of Hydrogen (and Helium to a lesser extent) and the overall neutrality of baryonic matter.

(b) By point 29, for a fixed dipole strength and nebula's geometry, $C$'s dependence on the number of dipoles, $n$, is concave, as larger $n$ imply larger negative-energy cross-terms per dipole in (28). This is clearly true also for the ZPF intensity inside $\mathcal{N}$ or in its immediate neighborhood. A natural such concave dependence would satisfy

$$\frac{\mathrm{d}C}{\mathrm{d}n} \propto \frac{1}{C} \quad \Rightarrow \quad C \propto \sqrt{n}. \tag{35}$$

To wit, the contribution of an added dipole to the halo energy density (30) is (statistically) suppressed by exactly the already present energy density interfering with it, which is proportional to $C$.

### 4.1.2. Spiral Galaxies

The best laboratories for testing dark-matter theories are spiral (or disk) galaxies. Matter in the disk's plane circles the galactic center at a speed which depends only on its distance from the galactic center—a dependence known as a *rotation curve*. This makes spirals the only astronomical objects in which the local *acceleration vector* of moving matter can be reliably inferred from the projection of their velocity on the line-of-sight, as deduced from the Doppler shift of their emitted spectral lines.

One universal feature of rotation curves concerns the radius beyond which they begin to significantly deviate from the Newtonian curve, viz. that which is calculated based on GR + baryonic matter [11]. This happens at around a radius at which the radial acceleration of orbiting matter equals some universal (small) value, $a_0$, known as the MOND acceleration. That such behavior, having no plausible explanation within the dark-matter paradigm, is expected from our model, followed by first asking: Where should "ZPF dark matter" kick-in? Point (29) implies that the energy content of the disc near its dense center is baryon-dominated until a critical radius, $R_c$, beyond which its surface density drops below some critical value $\Sigma_c$, and this conclusion is independent of the exact interference model which can only affect $\Sigma_c$'s value and the transition region, $R \sim R_c$. For the same reason, the halo's intensity inside a sphere of radius $R_c$, co-centered with the disc, is much smaller than implied by (30).

It follows that, under the approximation wherein the sphere's entire mass content is "moved" to its center, the rotation curve should be approximately Newtonian for $R \lesssim R_c$. Now, given that the coarse grained surface density of any spiral can be fitted rather well using only two parameters

$$\Sigma(R) = \Sigma_0 e^{-R/R_d}, \tag{36}$$

it can then be shown by a straightforward calculation that the radial acceleration at $R_c$ takes the form $a = 2\pi G\Sigma_c \mathcal{F}(\Sigma_0/\Sigma_c)$ for a slowly varying function $\mathcal{F}(x)$, approximately equal to 1 on $x \in [2, 12]$ (as is the case for $x$ in most spirals). The MOND result follows upon defining $a_0 := 2\pi G\Sigma_c$.

Moving beyond $R_c$ in the galactic plane, the rotation curve eventually flattens, as follows from the asymptotic gradient, (34) (the contributions of baryonic matter and of small $|x|$ corrections to (30), both drop faster, as $R^{-2}$). The square of that asymptotic (tangential) speed correlates rather well with $a_0$ and the total mass, $M$, of the galaxy and reads $\sqrt{GMa_0}$. This relation, also known as the Baryonic Tully–Fisher relation (BTFR), is also compatible with our model simply on dimensional grounds. As $\Phi$ is a solution of (24), the $C$ appearing in (34) must have dimensions $[C] = \text{mass}/\text{length}$. We further want it to monotonically increase with $R_d$ and with $\Sigma_0$—both increasing with the number of radiating dipoles. In the latter case, $C$'s dependence should be concave, as more particles also imply greater cross-term destructive interference. Finally, $C$ should monotonically increase also with $\Sigma_c$. A larger $\Sigma_c$ implies smaller interference effects, meaning that more radiation escapes the galaxy (note, again that $\Sigma_c$ already incorporates the details of any reasonable interference model). The obvious candidate up to a dimensionless coefficient is $C \propto \sqrt{\Sigma_0 R_d^2 \Sigma_c} \propto \sqrt{M\Sigma_c}$, rendering the full coefficient of the gradient (34) $\sqrt{G^2 M\Sigma_c} \propto \sqrt{GMa_0}$, which is the BTFR. Note that, as $M$ is proportional to $n$, the number of dipoles contributing to $C$, the above choice conforms with (35). The dimensionless coefficient can only be a function of the ratio $\Sigma_c/\Sigma_0$. The above conditions on both $\Sigma$'s imply that this function is slowly varying and, moreover, of an argument belonging to a relatively short interval.

An apparent weakness in the above analysis is that MOND's independence on the spiral's composition is not readily explained. It is not entirely clear why, e.g., a frenetic electron in a star's core should, on average, contribute to the ZPF on par with a bound hydrogen electron in a gas-dominated spiral, given their vastly different bulk motions. This composition independence suggests that it is not directly bulk acceleration which is responsible for the ZPF but, instead, the accompanying fluctuations in an electron's morphology and associated $\alpha$. In other words, the ZPF is a manifestation of energy exchange between electrons which are temporarily out of scale equilibrium. In light of the properties of the ZPF discussed in Section 2.3, it is rather natural for all electrons actively participating in such redistribution of energy to contribute, on average, similar amounts.

Summarizing, the MOND phenomenology, attributing a fundamental significance to $a_0$, appears as a deceiving coincidence between the relatively large ZPF contribution to the mass in sparse regions of a galaxy and the small Newtonian acceleration there. Besides in spirals, this coincidence manifests itself also in dwarf spheroidals and other, low-density pressure-supported systems.

### 4.1.3. Clusters of Galaxies

When dealing with the dynamics of galaxies in a cluster, the upper cutoff $r_1$ in (32) can no longer be arbitrary and must represent the physical radius, around which a galaxy's ZPF halo merges with the Friedman ZPF DC, rendering the galaxy's mass finite. In sufficiently sparse clusters, the halo size can reasonably be assumed to be much smaller than the average intergalactic distance; hence, in the Newtonian approximation, each galaxy can be modeled by a point in which *gravitational* mass comes mainly from its halo.

What about a galaxy's *inertial* mass? The geodesic equation, even in its Newtonian approximation (25), is of course oblivious to this question, and a galaxy should move in the field of all others independently of that extra ZPF energy it carries. Nonetheless, if we wish to make full use of Newtonian gravity and, in particular, its expression for the conserved e-m, the halo must be treated as

if also contributing to a galaxy's inertial mass. When doing so, the virial theorem is a legitimate tool for estimating the contribution of ZPF halos to a cluster's mass (The alert reader may ponder whether this extra inertial mass is consequential to electromagnetic interaction between particles. The answer is negative, as is evident from the derivation of the Lorenz force equation in appendix D of [1]).

The flat rotation curve of spirals typically persists to the observational limit, which could be at ten times a galaxy's optical diameter or even beyond. It is therefore unclear which clusters qualify as "sufficiently sparse" and, consequently, whether the virial theorem is a legitimate tool for estimating their mass. From the above lower bound on a galaxy's halo size, it is however clear that, in some clusters or at least in the central regions thereof, a typical halo size certainly exceeds intergalactic separation distance. In such cases, two complications arise. First, as the halos are typically not spherically symmetric, the point-mass approximation collapses. Second, the hitherto ignored interference cross-terms between dipoles belonging to distinct galaxies must be taken into consideration. Specifically, as relativistic e-m must be conserved, the electromagnetic e-m subtracted from two halos approaching each other as a result of such destructive interference must appear elsewhere, such as in the bulk motion of galaxies or in the intracluster medium.

There is, nonetheless, a good indication that naively "attaching a fixed halo" to each galaxy is a decent first approximation. Most of a cluster's baryonic mass (90–80%) comes from the Intracluster Medium (ICM)—a tenuous plasma of average mean-free-path $\sim$ 1 light-year. The gravitational potential in the cluster can be inferred from the density and temperature distributions of the ICM [12]. From that potential, one can determine (via Poisson's equation) the sourcing mass distribution which, as predicted, is found to be dominated by some "phantom mass" for which the distribution follows rather closely that of galaxies rather than of the ICM. As a bonus, this result provides a confirmation for our model's prediction that tenuous plasma contributes only little to the ZPF.

### 4.1.4. Gravitaional Lensing

An apparent independent confirmation for the last point above, comes from the so-called "Bullet Cluster" (1E 0657-558), whose alleged collision with another cluster had stripped both from their ICM, leaving two clusters composed virtually of galaxies only. Although the mass of the plasma left behind greatly exceeds that of the bare clusters, the total mass distribution in the region of collision, as inferred from weak gravitational lensing of background galaxies, is dominated by phantom mass whose distribution correlates well with the distribution of galaxies alone [13] (Abell 2744—Pandora's Cluster—is yet another good example). Nevertheless, our putative electromagnetic dark matter differs from ordinary cold dark matter in many respects. Inferring $p_{00}$ from lensed images of background objects might yield erroneous results if our model is in fact valid—hence a questionable confirmation.

Conventional gravitational lensing calculations are all based on null geodesics in the following degenerate metric:

$$(\mathrm{d}s)^2 \equiv (1 + h_{00})(\mathrm{d}t)^2 + (\eta_{ij} + h_{ij})\mathrm{d}x^i\mathrm{d}x^j = (1 + h_{00})(\mathrm{d}t)^2 + (-1 + h_{00})|\mathrm{d}\boldsymbol{x}|^2, \tag{37}$$

where $h_{00}$ is sourced by $p_{00}$ of both dark and ordinary, nonrelativistic matter. The source $p_{00}$ cannot be uniquely deduced from the image, and various additional assumptions are needed for that, e.g., spherical DM halo in the case of single galaxies. This is one place where our model departs from CDM (cf. Section 4.1.1). Even more critically, in our model $p_{ij} \neq 0$ for the dark-matter part of the e-m tensor and, consequently, $h_{ij} \not\propto \delta_{ij}$, invalidating (37).

It might appear that the addition of six extra "dark components", $p_{ij}, i \geq j$, to the (symmetric) e-m tensor has rendered our model too flexible to be refuted by any observation, but this impression is wrong. From $p_\mu{}^\mu = 0$, the trace of the so-called Maxwell stress, $p_{ij}$, must equal $p_{00}$, while $\partial_\mu p^{\mu\nu} = 0$ and $\partial_0 p^{0i} = 0$ implies $\partial_j p^{ji} = 0$. Moreover, exactly the same destructive interference affecting $p_{00}$ (Section 4.1.1) also affects $p_{ij}$ and, given a detailed (reasonable) interference model, $p_{ij}$ would essentially be uniquely determined by $p_{00}$. For example, the nebula's halo analyzed in Section 4.1.1

must have $p_{ij} \sim p_{00}\hat{x}_i\hat{x}_j$ for large $|\pmb{x}|$, on the above grounds only, implying $h_{ij} \sim h_{00}\hat{x}_i\hat{x}_j$. The deflection angle of a light-ray traveling in the outskirts of a spherically symmetric halo is 25% smaller in our model than in a CDM (isothermal) halo having the same effective $p_{00}$ (recall the factor 2 relative to (24)). Gravitational lensing, especially strong lensing by non-isotropic halos, is therefore an excellent laboratory for confronting our model with standard CDM.

## 5. Conclusions

Cold dark matter as a solution to the dark-matter problem suffers from two major drawbacks: it must be extremely esoteric to keep defying direct detection for so long, and it seems to "conspire" with baryonic matter in an inexplicable way. Our proposal that the missing component of the energy-momentum tensor is due to variations in some ZPF due to baryonic matter suffers from neither problem.

The proposed "transparent matter" model must obviously be advanced further for it to produce accurate predictions, but even in its preliminary form, it makes directly testable qualitative predictions: (a) Tenuous plasma generates very little such transparent matter, compared with neutral gas and stars. (b) Relativistic motion can probe nonvanishing components of the Maxwell stress, absent in cold dark-matter (Section 4.1.4 above). (c) Disc galaxies have a non-spherically symmetric halo, declining as $f(\hat{\pmb{x}})|\pmb{x}|^{-2}$ for large $|\pmb{x}|$, with $f(\hat{\pmb{x}})$ minimal in the galactic plane. (d) The success of the MOND phenomenology in single galaxies is explained (as a mere coincidence) as is its failure in clusters.

The existence of such a ZPF and its unique properties (distinguishing it from the SED ZPF) have been inferred in previous work by the author, which rendered classical electrodynamics well defined—ECD. That work also implies that QM can be viewed as that necessary, complementary theory, dealing with statistics of ensembles of ECD solutions, while the current work further gives support to the possibility that nothing beyond ECD is needed to describe all forms of matter, offering simple explanations to persistent mysteries in particle physics, such as the quantization of the electric charge and the observed particle - antiparticle imbalance. In both cases, it is the tiny masses of particles, rendering the ZPF consequential to their behavior, despite its ultra low-intensity; In astrophysics it is the huge volumes involved. Since for many decades now, an unprecedentedly large, well informed and massively funded community, has been making no progress in all those foundational question, it is proposed that a single wrong turn at the beginning of the twentieth century, is the root-cause for this long stagnation, and that dark matter is yet another ignored facet of good-old—though properly fixed—CED.

**Funding:** This research received no external funding.

**Conflicts of Interest:** The author declares no conflict of interest.

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
