# Peer review of "Dark Matter as Variations in the Electromagnetic Zero-Point Field Induced by Baryonic Matter"

_symmetry, doi:10.3390/sym12091534_

Round 1

Reviewer 1 Report

In point 1 of the introduction "Freely moving charges in a lab, trace parabolas rather than straight lines" needs to mention gravity. Eg it could be written "Charges moving freely under gravity in a lab, trace parabolas rather than straight lines". I think it is a little exaggerated to say it is "dead wrong", I would leave that first sentence out, without necessarily taking away from the thrust of the paragraph.

In point 2 more explanation as to why there is a self-force problem is needed than just the reference, particularly since it is to the author's own paper, which does not fully explain the problem. A reference to a paper that explains more fully what the self-force problem is would be useful. Points 3 and 4 should also have reference(s), and some examples of what the "discrepancies" are.

page 2 "As it turns out, such proper fixing is indeed possible." should have a reference.

page 16 line 4 should be field not filed

page 19 sentence starting on the 3rd last line needs to be rephrased

Reference [15] needs to be accessible. If not yet published, a copy should be on the archives before this work is published.

I would prefer to see much more detail in the theory itself, including a a more simplified and more detailed explanation of the physical basis of ECD and its relation to quantum mechanics. 

Author Response

Reply regarding:

Paragraph 1. Point well taken.

Paragraph 2.  I truly think that my ref. [2] gives the best explanation of what the self-force problem is, but I added a ref. to Jackson's book (confessing that "...the self-force problem has never been really solved").

Broken DOI to [15] fixed.

Point 3 in the introduction is my original (I think) take on general covariance, hence I can't give references. It should be self-explanatory to anyone with the necessary mathematical background.  

The reviewer wishes to see a more detailed explanation of ECD's physical basis. As I argue in the introduction, ECD is just well defined classical electrodynamics (whose physical basis is undisputed). ECD's relation to QM, i.e., that of (well defined) classical electrodynamics, is discussed at length in [4]. I can repeat some of it in the current paper but at the cost of increasing the size of the paper. 

Reviewer 2 Report

This paper is very long and quite difficult to understand since the author often engages in long digressions about several aspects of 20th century physics. If any, I don't see what are the original scientific contributions of this paper. As stated in the manuscript, Sec. 2 is a summary of previous work by the author himself and Sec. 3 is "not crucial for the understanding of the rest of the paper, and may be skipped on first reading". Finally, in Sec. 4, the author refers several times to reference [15] where it seems that the cosmological results of the theory will be presented.

For the above reasons, unfortunately I can not recommend this paper for publication.

Author Response

I have made several changes and additions to the new draft.

I hope that the newly written abstract and title, in combination with the added conclusion section, make it clear "what the original scientific contributions of this paper are", and added section 4.1.4 points to direct tests distinguishing between cold dark matter and my putative `electromagnetic dark matter'.

The new abstract also explains in what respect my proposed `transparent matter' is greatly superior to cold dark matter and I dare say that no physicist would object to that conclusion if convinced that it is compatible with well-tested terrestrial physics (added paragraph 3 to sect. 4.1 elaborates on this point). In other words, my proposed `dark matter', is not yet another placeholder for `something' with more non vanishing components for its energy-momentum tensor, than cold dark matter; Its possible incompatibility with particle physics can't be explained away by "extensions", "higher energy", "new sectors" etc. of standard particle physics. In particular, both advanced and retarded electromagnetic fields must play a role in it, which is therefore intimately linked with the classical self-force problem (add quote from Jackson – footnote 1 – is illuminating in this regard). 

 It is for this reason that sect. 3 has been included in the paper, while sect. 2 is the absolute minimum required to grasp a universe where both retarded and advanced fields play an essential role (as is sect. 3.4.1 which is referred to in section 4). I'll be happy to further motivate their inclusion, beyond what is already in the introduction.

Broken DOI to a preprint of [15] has been fixed. My apologies.    

Reviewer 3 Report

This is an interesting manuscript in which the author argues that the quantum vacuum (if I understood this correctly) needs to be taken into account in the description of elementary particles and that the description proposed might account for the observed deviant dynamical behaviour on the scales of galaxies and galaxy clusters. The general gist is indeed quite exciting but the manuscript appears to be a mixture of a philisophical self-debate with little actual quantification of the issues at hand. 

A few comments which might hopefully be useful to the author:

The reference [1] is not in the text. There are far too many acronyms and I had to spend tooo much time to try to find the definitions. A table of these would be useful. Some do not appear to be defined, e.g. EM or e-m (what is "e-m conservation"?). Many statements appear to come from the blue. E.g. "The most general dependence" just above Eq.10 - perhaps this is clear to the expert? Another e.g.:  "The standard result treating the metric perturbation as a scale-scalar, entails the appearance of two non-scalars in linearized Einstein's equation". What is "specie" on the top of p.9 (occurs two times)? What is an "internal current in a scalar ECD solution" (p. 11 in Sec. 3.1.1)?

At the beginning of Sec.3 there are many words. In general it is unclear how quantitative this approach is? How can the shape of the rotation curve of a galaxy be calculated based on the present suggestions?  What is a "retarded "EM pulse" (p.14)? 

The way the dark matter effect appears is not really explained well. Past authors (e.g. Smolin, Verlinde, also Milgrom) had already suggested that the Milgromian constant a_o may be an effect of the quantum vacuum. Why is this ignored here? The author refers to MOND being an ugly theory with an infinite number of tunable parameters (p.19), but the author fails to show how the baryonic Tully-Fisher relation (BTFR) comes out of the ECD theory (it is a direct and simple implication from Milgrom's theory). In this line of thought, the assumed exponential surface density distribution (Eq.34) needs to be a result of a gravitational theory, rather than being the input into it.

The present manuscript does not seem to make clear how MOND can be replaced by this supposedly more fundamental description, MOND indeed being the bench-mark since in this _simple_ description basically all known astronomical phenomena come out correctly. It is important to search for a more fundamental theory which encompasses MOND, but it is not clear how the present theory would achieve this as the generalised MOND Poisson equation, for example, is not shown to emerge.

Author Response

The reviewer appears to be an enthusiast of MOND, so let me start with a personal anecdote. As a grad student at Weizmann, I had the privilege of discussing MOND with Milgrom. In response to my criticism that, compared with GR (which he was teaching us at that time) MOND seems out of the blue, lacking any physical motivation, he replied: "Tell that to Schrodinger too".  

It took me several years, but now I can show that Schrodinger can't be told so; his equation is just the simplest way to statistically describe classical electrodynamics [4] - provided the latter is given a consistent definition, which is what ECD is all about. 

On a more conceptual level I ask, along with some DM advocates: What is the point in astronomy and cosmology if they can't be intimately linked with terrestrial scale physics? Those two disciplines will forever remain limited to the passive collection of electromagnetic (and perhaps gravitational) waves, from a single point-of-view. A deformation of terrestrial physics - surviving a plethora of active tests - which is experimentally inaccessible otherwise, is not really different from epicycles drawing, and will always be achieved with enough of them - interpolations functions, extra fields and terms in the Lagrangian etc. (I have add this paragraph to the beginning of sect. 4.1.1 ). 

It should therefore be clear (and I hope I have made it so in the newly written abstract) that my proposal is not taking part in the how-many-tunable-parameters contest,  typical of non-DM models. Instead, it argues for a hitherto ignored link between the micro and macroscopic world. Consequently,  the behavior of that missing (transparent) component of the energy-momentum (e-m) tensor is very complicated, involving among else the phase of matter in its vicinity. I suspect that a full-fledged model would not even come close the analytic compactness of MOND and the like.   So, while it's true that my derivation of the BTFR is currently more heuristic than MOND's (eq. (34) adds yet another such heuristic support) I give strong arguments for the compatibility of my model with observations involving clusters (the Bullet cluster in particular) - a category where MOND fails terribly (the reviewer's statement that "...all known astronomical phenomena come out correctly [in MOND]" is very odd). Moreover, BTFR is merely a statistical property of typical galaxies, with a scatter and outliers. On the other hand, the flattening itself of the rotation curve is true for every (well resolved) galaxy, and this property emerges inevitably from Maxwell's equations, without any ad hoc assumptions, as in MOND. I have also added sect. 4.1.4 which gives a direct test discriminating between cold dark matter and the putative `electromagnetic dark matter' . A conclusion section has also been added.

As for the exponential profile (34) - surely the reviewer doesn't expect me to simulate the evolution of galaxies within my proposal in order to show that (34) is indeed attained - an endeavor which only recently has been attempted within MOND. If so, then I have no idea what is meant by "the assumed exponential surface density distribution (Eq.34) needs to be a result of a gravitational theory"; I'm using well-tested GR all along.

Finally, I'm well aware of that works of  Smolin, Verlinde and Milgrom, but I fail to see their direct relevance to the current paper. At no point do I mention, let alone analyze the properties of the "quantum vacuum", but rather that of some `zero point field', unique to ECD, which is allegedly behind most quantum phenomena, as explained in [4].

Acronyms etc. have been taken care of. 

Round 2

Reviewer 2 Report

The author has performed some cosmetic changes, but at least he has written a Conclusions section, which was absent in the previous version. The abstract has also been improved.

I think that the transparent matter model that is presented in this paper might point to an interesting research line. Even though, I think that in this manuscript the model is not very clearly explained and many details are still to be worked out in order to obtain a properly defined model with clear cosmological predictions.